# Mitochondrial AOX Supports Redox Balance of Photosynthetic Electron Transport, Primary Metabolite Balance, and Growth in *Arabidopsis thaliana* under High Light

**DOI:** 10.3390/ijms20123067

**Published:** 2019-06-23

**Authors:** Zhenxiang Jiang, Chihiro K. A. Watanabe, Atsuko Miyagi, Maki Kawai-Yamada, Ichiro Terashima, Ko Noguchi

**Affiliations:** 1Department of Biological Sciences, Graduate School of Science, The University of Tokyo, Bunkyo-ku, Tokyo 113-0033, Japan; kyoshinsho@gmail.com (Z.J.); chihiro0223@gmail.com (C.K.A.W.); itera@bs.s.u-tokyo.ac.jp (I.T.); 2Graduate School of Science and Engineering, Saitama University, 225 Shimo-Okubo, Sakura-ku, Saitama-city, Saitama 338-8570, Japan; miyagi@mail.saitama-u.ac.jp (A.M.); mkawai@mail.saitama-u.ac.jp (M.K.-Y.); 3School of Life Sciences, Tokyo University of Pharmacy and Life Sciences, 1432-1 Horinouchi, Hachioji, Tokyo 192-0392, Japan

**Keywords:** alternative oxidase, metabolic interaction, mitochondrial respiratory chain, photosynthesis, redox balance

## Abstract

When leaves receive excess light energy, excess reductants accumulate in chloroplasts. It is suggested that some of the reductants are oxidized by the mitochondrial respiratory chain. Alternative oxidase (AOX), a non-energy conserving terminal oxidase, was upregulated in the photosynthetic mutant of *Arabidopsis thaliana*, *pgr5*, which accumulated reductants in chloroplast stroma. AOX is suggested to have an important role in dissipating reductants under high light (HL) conditions, but its physiological importance and underlying mechanisms are not yet known. Here, we compared wild-type (WT), *pgr5*, and a double mutant of AOX1a-knockout plant (*aox1a*) and *pgr5* (*aox1a/pgr5*) grown under high- and low-light conditions, and conducted physiological analyses. The net assimilation rate (*NAR*) was lower in *aox1a/pgr5* than that in the other genotypes at the early growth stage, while the leaf area ratio was higher in *aox1a/pgr5*. We assessed detailed mechanisms in relation to *NAR*. In *aox1a/pgr5*, photosystem II parameters decreased under HL, whereas respiratory O_2_ uptake rates increased. Some intermediates in the tricarboxylic acid (TCA) cycle and Calvin cycle decreased in *aox1a/pgr5*, whereas γ-aminobutyric acid (GABA) and N-rich amino acids increased in *aox1a/pgr5*. Under HL, AOX may have an important role in dissipating excess reductants to prevent the reduction of photosynthetic electron transport and imbalance in primary metabolite levels.

## 1. Introduction

In illuminated leaves, the function of the mitochondrial respiratory chain should be essential for maintenance of the photosynthesis system [1,2,3]. For example, the respiratory chain functions in the oxidation of excess reductants produced in chloroplasts, the oxidation of NADH produced from the photorespiratory glycine decarboxylase complex, and ATP supply for cellular processes (e.g., cytosolic synthesis of sucrose) [4,5]. The plant mitochondrial respiratory chain consists of components that are associated with ATP production such as complex I, complex III, and cytochrome *c* oxidase (COX), and bypass pathways that are not associated with ATP production such as type II NAD(P)H dehydrogenases (NDs) and alternative oxidase (AOX) [6,7]. These bypass pathways are energetically wasteful. Many previous studies have reported that AOX is often induced under various stress conditions such as low temperature, high light (HL), and drought, and genes of NDs and AOX are coordinately expressed under stress conditions [8,9].

AOX1a knockout (AOX1a-KO) plants (*aox1a*) of *Arabidopsis thaliana* show very small amounts of AOX in their leaves, but their growth is not different from that of wild-type (WT) [10,11]. However, in *aox1a* during short-term HL stress, the plastoquinone (PQ) pool of photosynthetic electron transport was more reduced and the electron transport rate of photosystem II (PSII) under strong actinic light (AL) intensity was lower than in WT [12,13]. AOX-antisense plants suffered from more severe PSII photoinhibition after HL treatment compared to WT and AOX-overexpressed plants [14]. In the presence of a complex III inhibitor, *aox1a* was subjected to more severe PSII photoinhibition than WT. In particular, the PSII repair process was more damaged in *aox1a* in the presence of the complex III inhibitor [15]. Under moderate drought stress, the electron transport of PSII was lowered and protons accumulated in the thylakoid lumen in AOX-knockdown (AOX-KD) tobacco leaves. In AOX-KD tobacco, the operating efficiency of PSI (YI) was not inhibited and the cyclic electron flow around photosystem I (CEF-PSI) was upregulated [16,17]. In previous studies, AOX was considered to function by dissipating excess reductants under HL or drought stress conditions, where excess reductants accumulate in the chloroplasts and mitochondria. Unless AOX functions, reactive oxygen species (ROS) may be produced in the chloroplasts, and ROS may attack the repair cycle of damaged PSII [15]. AOX has also been suggested to have another function in cellular metabolic balance. In previous studies, *aox1a* showed changes in metabolite patterns compared to WT. In particular, levels of some amino acids, organic acids, and sugar phosphates markedly changed under stress conditions [11,18]. However, underlying mechanisms affecting changes in metabolic profiles in *aox1a* have been unsolved.

On the other hand, photosynthetic mutations affect the mitochondrial respiratory chain. AOX is induced under HL in *Arabidopsis* leaf-variegated mutant *yellow variegated 2* (*var2*) compared to WT [19]. The *var2* mutants lack FtsH2 metalloprotease, which is involved in the degradation of the D1 protein of PSII in the PSII repair cycle [20]. AOX is also induced under HL in Arabidopsis *proton gradient reduction 5* (*pgr5*) compared to WT [21]. In *pgr5*, the acceptor side of PSI is inhibited, excess reductants accumulate in chloroplast stroma, and PQ is reduced [22,23,24,25]. In the double mutant of *pgr5* and *chlororespiratory reduction 4-3* (*crr4-3*), which is deficient in the NAD(P)H dehydrogenase (NDH)-dependent pathway, the AOX protein amount and KCN-resistant respiratory rate were much higher than in WT even under low-light (LL) growth conditions, although *pgr5* and *pgr5/crr4-3* did not change the electron flow through AOX after HL treatment [26]. In a study of the physiological function of increased AOX in *pgr5* [13], the double mutant of *aox1a* and *pgr5* (*aox1a/pgr5*) grown under LL showed photosynthetic traits and growth similar to WT and *pgr5*. However, *aox1a/pgr5* grown under HL showed inhibited growth compared to WT and *pgr5*. The increased level of AOX in *pgr5* may have a function in growth under HL, but it has not been clarified what mechanisms affect the slower growth of *aox1a/pgr5* under HL and at what growth stage the AOX1a deficit intensively affects plant growth.

In this study, we aimed to clarify the following questions: (1) Does the mitochondrial AOX efficiently function in oxidation of excess reductants that accumulate under HL, especially in the photosynthetic mutant? (2) At what growth stage does AOX intensively affect plant growth under HL? (3) How does AOX affect the balance in cellular primary metabolites under HL? In order to solve these questions, we paid attention to *A. thaliana pgr5*, which accumulates excess reductants in chloroplasts and induces AOX under HL. We used the double mutant *aox1a/pgr5* and compared its growth traits and physiological characteristics with those of WT and *pgr5*. From the early growth stage, we conducted growth analysis and photosynthetic and respiratory measurements in WT, *pgr5*, and *aox1a/pgr5*. To elucidate the differences in growth and photosynthetic/respiratory traits, we measured expression levels of some mitochondrial respiratory chain genes, the redox state of PQ and ubiquinone (UQ), and primary metabolite levels in leaves. Since *pgr5* and *aox1a* often suffer from HL damage, we used plants grown under LL (LL plants) and HL (HL plants) conditions. The results indicate that *aox1a/pgr5* showed slow growth under both HL and LL, that the difference in growth can be ascribed to physiological parameters related to photosynthesis and respiration, and that PSII parameters and the balance of primary metabolite levels are affected in *aox1a/pgr5*.

## 2. Results

### 2.1. Low Net Assimilation Rate in aox1a/pgr5

In HL plants, we examined the total dry weight (DW) from 1 to 5 weeks after germination and total leaf area from 2 to 4 weeks after germination. Total DW and total leaf area were significantly different among genotypes (Figure 1A,B). At 2 and 3 weeks after germination, these values of *aox1a/pgr5* were significantly lower than that of *pgr5* (*p* < 0.05, except for leaf area at 3 weeks after germination, *p* = 0.0713). We estimated relative growth rate (*RGR*) from total DW (Figure 1C). *RGR* was not significantly different among genotypes (*p* > 0.1, analysis of covariance (ANCOVA)). We decomposed *RGR* into the morphological parameter, leaf area ratio (*LAR*), and the physiological parameter, net assimilation rate (*NAR*). *LAR* was significantly different among genotypes, and *LAR* was higher in *aox1a/pgr5* than in *pgr5* at 3 weeks after germination (*p* = 0.0700; Figure 1D). Furthermore, *LAR* can be decomposed into specific leaf area (*SLA*), which denotes leaf thinness, and leaf weight ratio (*LWR*), which denotes the ratio of leaf weight to total weight. *LWR* was similar among genotypes (data not shown), but *SLA* was significantly different among genotypes (Figure 1F). *SLA* of *aox1a/pgr5* was higher than that of *pgr5*. *NAR* was also significantly different among genotypes (Figure 1E). In contrast to *LAR* and *SLA*, *NAR* of *aox1a/pgr5* was lower than that of *pgr5* (*p* = 0.0612 at 2 weeks after germination). *NAR* is often correlated with net daily carbon gain, which is determined by whole plant photosynthesis and respiration rates [27]. Therefore, the lowest *NAR* of *aox1a/pgr5* may be due to the low rate of photosynthesis of this double mutant. 

We also compared parameters of growth analysis of LL plants. The growth rate of LL plants was slower than that of HL plants, thus we examined total DW from 1 to 6 weeks and total leaf area from 2 to 5 weeks (Figure 2A,B). In LL plants, total DW and total leaf area were significantly different among genotypes (Figure 2A,B; *p* < 0.05, except for 1 week of total DW, *p* = 0.0541). At any growth stage, these values of *aox1a/pgr5* were lower than those of *pgr5*. Similar to HL plants, *RGR* of LL plants was not different among genotypes (Figure 2C). In LL plants, the difference in *LAR* among genotypes was smaller than that in HL plants (Figure 2D). The difference in *SLA* in LL plants was also smaller than that in HL plants (Figure 2F). In contrast to *LAR*, *NAR* of *aox1a/pgr5* was lower than that of *pgr5* at 4 weeks after germination (Figure 2E).

### 2.2. PSII Parameters Are Lower in aox1a/pgr5 under High Actinic Light Intensity

The results of the growth parameter *NAR* suggest the possibility that leaf photosynthesis rates are different among genotypes at the early growth stage. Since it is difficult to measure the CO_2_ assimilation rate of a small leaf at the early growth stage, we conducted chlorophyll fluorescence analysis of whole shoots or leaves and compared the parameters of PSII among genotypes. We used two AL intensities (40 and 245 µmol·m^–2^·s^–1^) close to growth light intensities and one strong AL intensity (606 µmol·m^–2^·s^–1^). In HL plants at 11 days after germination (11-day HL plants), the operating efficiency of PSII (YII) was different among genotypes at 245 and 606 µmol·m^–2^·s^–1^, similar to *NAR* (Figure 3A). The difference among genotypes was larger under higher AL intensity. Although the difference in YII between *pgr5* and *aox1a/pgr5* was insignificant, YII of *aox1a/pgr5* was lower than that of *pgr5* (Figure 3A). YII was decomposed into Fv’/Fm’, PSII operating efficiency of open PSII reaction center, and photochemical quenching (qP), the ratio of open PSII reaction center. qP was significantly different among genotypes (Figure 3B), thus the photosynthetic electron transport was more reduced in the mutants. Non-photochemical quenching (NPQ) was highest in WT (Figure 2C). In a previous study [22], NPQ of *pgr5* was reported to be low. NPQ of *aox1a/pgr5* was also low, similar to that of *pgr5*. Fv/Fm, the maximal efficiency of PSII, was not different among genotypes, thus PSII was not photoinhibited in 11-day HL plants. In leaves of HL plants at 21 days after germination (21-day HL plants), PSII parameters were similar to those in shoots of 11-day HL plants (Appendix A), but Fv/Fm was low in *pgr5* and *aox1a/pgr5* compared to WT (Table 1).

In shoots of LL plants at 15 days after germination (15-day LL plants), YII was significantly different among genotypes (Figure 3D). The low value of YII in *aox1a/pgr5* was ascribed to the low value of qP under high AL intensity. The qP of *aox1a/pgr5* was lower than that of *pgr5* (*p* = 0.0551; Figure 3E). NPQ values of *pgr5* and *aox1a/pgr5* were similar and significantly lower than that of WT (Figure 3F). In LL plants at 30 days after germination (30-day LL plants), the PSII parameters were similar between *pgr5* and *aox1a/pgr5*, and significantly lower than those of WT (Appendix A).

### 2.3. Cyanide-Resistant Respiratory O_2_ Uptake Rate Is Upregulated in pgr5 Mutant, Not in aox1a/pgr5

Since *NAR* is correlated with the net daily carbon gain, which is the difference between photosynthesis and respiration, we compared leaf respiration rates on a fresh weight basis. In 11-day HL plants, the O_2_ uptake rate in the absence of KCN (total respiration rate, total R) was different among genotypes (*p* = 0.0534), and total R of *aox1a/pgr5* was highest (Figure 4A). We also measured O_2_ uptake in the presence of KCN (CN-resistant respiration rate, CN R), and estimated a ratio of CN R to total R (CN R/Total R). CN R and CN R/Total R were lowest in *aox1a/pgr5* and highest in *pgr5* (Figure 4B,C). These results of *pgr5* were similar to those in previous studies [21,26], in which the AOX protein amount and CN R in *pgr5* were high compared to those in WT. In 21-day HL plants, CN R and CN R/Total R were lowest in *aox1a/pgr5* and highest in WT (Appendix A). In aged leaf tissue, AOX may be induced even in WT under HL, similar to *pgr5*.

In LL plants, total R was not different among genotypes (Figure 4D, Appendix A). In contrast, CN R and CN R/Total R were higher in *pgr5* than in the others (Figure 4E,F, Appendix A). Even under LL conditions, AOX may be induced in *pgr5*. In contrast, both values were low in WT of LL plants.

### 2.4. PQ Is Reduced in aox1a/pgr5 under HL 

Yoshida et al. [12] reported that PQ and UQ reduction levels were significantly different between WT and *aox1a* during short-tem HL treatment. Although the ratio of reduced form to total amount of UQ (UQ reduction level) of 11-day and 21-day HL plants was not different among genotypes (Figure 5C, Appendix A), the ratio of reduced form to total amount of PQ (PQ reduction level) of 11-day HL plants was different among genotypes (*p* = 0.0627; Figure 5A). The PQ reduction level of *aox1a/pgr5* was higher than that of WT and *pgr5*. The total amount of reduced and oxidized PQ (PQ total) and of reduced and oxidized UQ (UQ total) were also significantly different among genotypes. These amounts were higher in *pgr5* and *aox1a/pgr5*. In 21-day HL plants, PQ reduction level was similar among genotypes (Appendix A), but both PQ and UQ totals were different among genotypes. The PQ and UQ totals of *aox1a/pgr5* was highest among genotypes (Appendix A).

### 2.5. Gene Expression of the Respiratory Chain Is Partly Changed in aox1a/pgr5

Gene expression levels of two genes of NDs, *NDA1* and *NDB2*, and one gene of uncoupling protein, *UCP1*, were upregulated in *aox1a* leaves after 4 °C treatment compared to WT [10]. These bypasses may contribute to the increase in O_2_ uptake rate in *aox1a* at 4 °C compared to WT. Thus, we measured gene expression levels of *NDA1*, *NDB2*, *UCP1*, and one COX gene, *COX6b,* in 11-day HL plants and 15-day LL plants. In 11-day HL plants, the expression levels of these genes were not different among genotypes (Figure 6A–D), but the *NDA1* expression level of *aox1a/pgr5* was higher in 15-day LL plants (Dunnett test, *p* = 0.0603; Figure 6E). In contrast, the expression levels of *NDB2* and *UCP1* were lowest in *aox1a/pgr5* of LL plants (Figure 6F,H). These genes may not contribute to the increase in total R in *aox1a/pgr5*.

### 2.6. Primary Metabolite Levels Are Unbalanced in aox1a/pgr5

In a previous study [26], short-term treatment of HL induced changes in levels of many primary metabolites such as amino acids in *A. thaliana* leaves, and some metabolite levels were well correlated with leaf respiratory rates. In this study, we changed growth light intensity (i.e., long-term treatment), and many primary metabolites in HL plants were significantly higher than those in LL plants (Figure 7, Appendix A). Some metabolite levels were significantly different among genotypes (Figure 7). Although levels of fructose 6-phosphate (fructose 6-P) and glycerate 3-phosphate (3PGA) in glycolysis were significantly different among genotypes (Figure 7D,F), levels of many intermediates of glycolysis, glucose 1-phosphate (glucose 1-P), glucose 6-phosphate (glucose 6-P), fructose bisphosphate (FBP), dihydroxyacetone phosphate (DHAP), glyceraldehyde 3-phosphate (glyceraldehyde 3-P), phosphoenolpyruvate (PEP) and pyruvate, and alanine (Ala), which is derived from pyruvate, were not different among genotypes (Appendix A). In tricarboxylic acid (TCA)-cycle organic acids, levels of succinate and fumarate decreased but the level of isocitrate increased in *pgr5* and *aox1a/pgr5* (Figure 7A–C). Since levels of some TCA-cycle organic acids were different among genotypes, we compared levels of amino acids synthesized from the TCA-cycle organic acids. The level of γ-aminobutyric acid (GABA) was higher in HL plants than in LL plants, and the GABA level of *aox1a/pgr5* was higher than that of the other genotypes (Figure 7J). However, arginine (Arg), ornithine, citruline, and proline (Pro), which are also synthesized from Glu like GABA, were not different among genotypes (Appendix A). A level of glutamine (Gln), which has a high nitrogen (N)/carbon (C) ratio, was significantly different among genotypes, and the level in *aox1a/pgr5* was higher than in the other genotypes (Figure 7K). A level of asparagine (Asn), which has also a high N/C ratio, was significantly different among genotypes, and the levels in two mutants were higher than that in WT (Figure 7L). In contrast, levels of the branched-chain amino acids leucine (Leu), valine (Val), and isoleucine (Ile), which have a low N/C ratio, were not different among genotypes (Appendix A). Since the photosynthetic electron transport rate was low in *aox1a/pgr5*, the photosynthetic CO_2_ assimilation rate may be low. In such plants, the amino acids that have a high N/C ratio may accumulate in leaves. Levels of some amino acids of the aspartate (Asp) group (Asp, methionine (Met), and threonine (Thr)) were lower in *aox1a/pgr5* (Appendix A). In the synthetic pathway of the Asp group, the fluxes of pathways may be changed in *aox1a/pgr5* compared to WT. Ratios of primary metabolites were significantly different among genotypes. Both ratios of glutamine to glutamate and (Gln/Glu) and of malate to aspartate (Mal/Asp) were higher in *aox1a/pgr5* than in the others; (Figure 7M,N).

A level of 6-phosphogluconate, which is an intermediate of the oxidative pentose phosphate, was different among genotypes, and a level of glycolate, which is an intermediate of photorespiratory pathways, was higher in *aox1a/pgr5* than in the other genotypes (Figure 7E,I). In contrast, levels of ribulose 5-phosphate (Ru5P) and ribulose-1,5-bisphosphate (RuBP), which are intermediates of the Calvin cycle, were significantly different among genotypes, and the levels in *aox1a/pgr5* were lower (Figure 7G,H). In *aox1a/pgr5*, the low rates of photosynthetic electron transport may lead to low activity of the Calvin cycle and perturb other pathways such as the photorespiratory pathway.

## 3. Discussion

### 3.1. Low Net Assimilation Rate in aox1a/pgr5 at the Early Growth Stage

In our previous study [13], compared to WT and *pgr5*, *aox1a/pgr5* showed similar growth under LL, but inhibited growth under HL. However, it has not been clarified what mechanisms affect the inhibited growth, and what stage is critical for growth inhibition. In this study, we showed that *NAR* of *aox1a/pgr5* was lower at the early growth stage in both HL and LL plants (Figure 1 and Figure 2). On the other hand, *LAR* of *aox1a/pgr5* was highest in HL plants. The high *LAR* may complement the low *NAR* and repress the decrease in *RGR*. Such increased *LAR* was observed in another study in which *NAR* was decreased by a decrease in photosynthetic rate due to virus infection [28]. The high *LAR* of *aox1a/pgr5* was ascribed to high *SLA*. In *aox1a/pgr5*, whole-plant photosynthetic gain may be complemented by the increase in light capture with large and thin leaves. Many previous studies suggested that AOX does not affect plant growth [7] except for AOX-antisense *A. thaliana* at low-temperature stress [29]. This study clearly indicates that AOX is essential for optimal photosynthesis and growth under the condition where excess reductants accumulate in chloroplasts.

### 3.2. Changes in Photosynthetic Electron Transport and Respiratory Chain in aox1a/pgr5

In our previous study [13], the photosynthetic traits of *aox1a/pgr5* were not different from those of *pgr5* in both HL and LL plants. However, since mature leaves at the late vegetative growth stage were used in the previous study, no clear difference between *aox1a/pgr5* and *pgr5* was observed. In this study, we clearly showed that YII under high AL intensity was lower in *aox1a/pgr5* than in the other genotypes at the early growth stage of both HL and LL plants, and that the lower YII was mainly determined by the lower qP in *aox1a/pgr5*. In *aox1a/pgr5* at the early growth stage, the PQ pool is reduced and the PSII electron transport may be inhibited. These should affect the lower *NAR* in *aox1a/pgr5*. In AOX-KD tobacco leaves under moderate drought stress, an amount of the chloroplast proton-ATP synthase decreased, and proton accumulated in the thylakoid lumen. The proton accumulation led to the inhibition of the electron transport in the cytochrome *b_6_f* complex, and thereby the electron transport of PSII was also inhibited. In those tobacco leaves, CEF-PSI was upregulated and high NPQ was induced [16,17]. In contrast, NPQ of *aox1a/pgr5* was not induced even under high AL intensity. Since the electron transport is inhibited on the acceptor side of PSI in *pgr5* [24,30], CEF-PSI and NPQ could not be induced in either *pgr5* or *aox1a/pgr5*. The low value of NPQ and acceptor-side limitation of PSI may induce an increase in ROS production in *aox1a/pgr5.* ROS can affect the recovery process of D1 protein in PSII [31]. Although we could not explain the differences in Fv/Fm between 11-day and 21-day HL plants and between 15-day and 30-day LL plants, PSII in *aox1a/pgr5* was photoinhibited to a small extent. In *aox1a* under strong AL intensity, a larger extent of PSII photoinhibition was observed than in WT [15,32]. AOX may function as maintenance of PSII at HL.

In illuminated leaves of *pgr5*, ferredoxin (Fd) is reduced compared to WT [24]. At HL, Fd may be more reduced. Although underlying mechanisms of Fd reduction in *pgr5* has been controversial [24,25], the electron of Fd in *pgr5* should be partly transported via the malate valve and oxidized by the respiratory chain. In many photosynthetic organisms other than angiosperms, an alternative electron transport mediated by flavodiiron protein (FLV) functions as the electron sink in the acceptor side of PSI [33]. In leaves of angiosperms in the absence of FLV, the oxidation of excess reducing equivalents by the respiratory chain may be important. Unless this oxidation occurs, the acceptor side of PSI may be reduced leading to the ROS production.

PGR5 is associated with PGRL1 [34]. In *pgrl1* mutant of *Chlamydomonas* that has FLVs, the PQ redox status and YII were largely affected by the respiratory inhibition compared to those in WT [35]. Under low CO_2_ and HL conditions, this effect was more intensive. Under these conditions, not only FLV but also AOX were induced. In leaves of C3 plants under drought condition, stomata close and intercellular CO_2_ concentration (Ci) decreases. Low Ci increases NADPH consumed by photorespiration [4]. In the photorespiratory pathway, NADH is produced in the mitochondrial glycine decarboxylase complex. The photorespiratory NADH oxidation by the respiratory chain is important to prevent ROS production, leading to the maintenance of PSII [4,15,32]. Under drought conditions, the AOX inhibition induced the decrease in YII even at LL in wheat [36], tobacco [16] and *A. thaliana* [18]. In contrast, under high CO_2_ condition where photorespiration rate is low, the effect of respiratory inhibition should be low, especially that of the AOX inhibition. Under high CO_2_ condition, the expression of AOX genes was not changed [37].

In tobacco leaves, the AOX expression level was changed depending on the PQ redox level. The AOX expression level was induced under the condition where the PQ pool was reduced [38]. The expression of NADP-malate dehydrogenase that functions in the malate valve may be similar to that of AOX [21]. Although in *pgr5* the PQ pool is highly reduced and the AOX expression is induced, the electron flux via AOX in the dark was low in *pgr5* even after the HL treatment [26]. In the dark, Fd should be quickly oxidized by several mechanisms in chloroplasts. The induced AOX in *pgr5* may be used under illuminated condition.

The respiration rates (total R) of 11-day HL plants were higher in *aox1a/pgr5* than in the other genotypes (Figure 3A). In *aox1a/pgr5*, AOX was not induced under HL, thus the cytochrome pathway (CP) may be increased. Such a high respiratory rate may be related to lower *NAR* in *aox1a/pgr5*. Florez-Sarasa et al. [39] reported that both total respiratory rate and electron flow through CP decreased with increased plant age in *A. thaliana* WT, but the electron flow through AOX did not change throughout plant age. Even at the early growth stage, AOX may have an important function in plant growth.

### 3.3. Changes in Primary Metabolite Balance in aox1a/pgr5

Giraud et al. [18] and Watanabe et al. [11] reported that the balance in primary metabolite levels was changed in *aox1a* compared to WT under stress conditions. Florez-Sarasa et al. [26] showed that changes in primary metabolite levels after HL stress were different between *pgr5* and WT. In this study, the levels of some TCA-cycle organic acids were significantly different among genotypes (Figure 7, Appendix A). Although the 2-oxo glutarate (2-OG) level was not different among genotypes, GABA and Gln, which are synthesized from 2-OG, accumulated in *aox1a/pgr5*. These results suggest that the TCA-cycle flux may change in *aox1a/pgr5* compared to the other genotypes. The changes in TCA-cycle flux may be supported by the results of Gln/Glu. This ratio in *aox1a/pgr5* was significantly higher than in the others. The levels of the primary metabolites 6-phosphogluconate and glycolate were highest, and those of Ru5P and RuBP were lowest in *aox1a/pgr5*. The high Asp/Mal in *aox1a/pgr5* suggests that the flux of Mal/Asp shuttle between the cytosol and mitochondrion may be changed in *aox1a/pgr5*. These changes suggest that primary metabolic pathways, the Calvin cycle, the oxidative pentose phosphate pathway, and the photorespiratory pathway may be perturbed in *aox1a/pgr5*, and that the perturbation may be due to the reduction of photosynthetic electron transport, the low rate of photosynthesis, and the high rate of respiration in *aox1a/pgr5*. Such unbalanced C and N metabolism may partly affect the slower rate of growth in *aox1a/pgr5*. The values of PQ total and UQ total were higher in *pgr5* and *aox1a/pgr5* than in WT. PQ and UQ are derived from tyrosine (Tyr) and phenylalanine (Phe), respectively [40,41]. Tyr and Phe are also precursors of other secondary metabolites. In this study, the levels of both amino acids were not different among genotypes (Appendix A).

In this study, we compared physiological and biochemical traits among WT, *pgr5*, and *aox1a/pgr5*, and concluded the following: (1) The mitochondrial AOX may function in oxidation of excess cellular reductants that accumulate in the photosynthetic mutant *pgr5* under HL. Unless AOX can function under HL, the photosynthetic electron transport is reduced and inhibited, leading to low *NAR*. (2) Especially at the early growth stage, AOX intensively affects optimal photosynthesis and growth under both HL and LL conditions. (3) The balance of cellular primary metabolites is affected by the AOX deficit, which may be due to the decreased photosynthetic rates and increased respiratory rates. In the future, the underlying mechanisms of these phenomena occurring in *aox1a/pgr5* at the early growth stage should be clarified.

## 4. Materials and Methods 

### 4.1. Plant Materials and Growth Conditions

We used *Arabidopsis thaliana* wild-type (Col-*gl1*), *pgr5* mutant, and *aox1a/pgr5* double mutant. The background of the 2 mutants is Col-*gl1*. The *aox1a/pgr5* double mutant was generated by crossing *aox1a* (SALK_084897) and *pgr5* in a previous study [13]. Seeds were sown in Metro Mix 350 (Sun Gro Horticulture, Agawam, MA, USA) and vermiculite (G20; Nittai, Osaka, Japan) at a ratio of 1:1 (*v*/*v*) in 130 mL plastic pots (Tokai, Gifu, Japan). Plants were irrigated with a 1/1000 dilution of a mineral nutrient solution (Hyponex; Hyponex Japan, Osaka, Japan). Plants were grown in an air-conditioned chamber (LPH200; Nippon Medical and Chemical Instruments, Osaka, Japan) at 23 °C with 60% relative humidity. The day length was 10 h, and the photosynthetic active photon flux density was 240 µmol photons m^−2^·s^−1^ for high-light (HL) and 50 µmol photons m^−2^·s^−1^ for low-light (LL) conditions. 

### 4.2. Growth Analysis

After germination, we sampled whole plants every week. We cut into shoots and roots, scanned all leaves with a scanner, and calculated whole leaf area with ImageJ (ImageJ 1.48, National Institutes of Health, Bethesda, MD, USA). The samples were dried at 80 °C for 3 days and weighed. Firstly, relative growth rate (*RGR*, g·g^–1^·day^–1^) was calculated as the slope of linear regression of the natural logarithm of plant dry weight as a function of time using the data of 3 sequential samplings. For example, *RGR* at 2 weeks was estimated using data from 1 to 3 weeks.

*RGR* can be decomposed into leaf area ratio (*LAR*, m^2^·g^–1^) and net assimilation rate (*NAR*, g·m^–2^·day^–1^) as Equation (1). *LAR* can be further decomposed into specific leaf area (*SLA*, m^2^·g^–1^) and leaf weight ratio (*LWR*, g·g^–1^) as Equation (2).

(1)$$RGR=\frac{1}{W}\frac{dW}{dt}=\frac{LA}{W}\times \frac{1}{LA}\frac{dW}{dt}=LAR\times NAR$$(2)$$LAR\;=\;\frac{LA}{W}=\frac{LA}{LW}\times \frac{LW}{W}=SLA\times LWR$$
where *W* is the dry weight of whole plant (g), *LA* is the leaf area of whole plant (m^2^), and *LW* is the leaf dry weight of whole plant (g). In this study, *LAR*, *SLA*, and *LWR* were calculated with data of each week as the following equations:(3)$$LAR=\frac{LA}{W}$$

(4)$$SLA=\frac{LA}{LW}$$

(5)$$LWR=\frac{LW}{W}$$

Furthermore, *NAR* was calculated with estimated *RGR* and *LAR* as Equation (6):(6)$$NAR=\frac{RGR}{LAR}$$

### 4.3. Chlorophyll Fluorescence Measurements

Chlorophyll fluorescence of shoots of 11-day HL plants and 15-day LL plants and mature leaves of 21-day HL plants and 30-day LL plants was measured with a pulse amplitude modulation fluorometer (PAM-101, Waltz, Effeltrich, Germany). After dark incubation for 20–25 min, the minimum fluorescence level in the dark-adapted state (Fo) was measured. A saturating pulse (SP, 1 s, 5000 µmol photons m^–2^·s^–1^) was subsequently applied to determine the maximum fluorescence level in the dark-adapted state (Fm). Actinic light (AL) was provided at 3 intensities (40, 245, 606 µmol photons m^–2^·s^–1^). After fluorescence under AL reached a steady-state level (Fs), an SP was given to measure the maximum fluorescence level during AL illumination (Fm’). Chlorophyll fluorescence parameters were calculated as previously described [42]. The maximal efficiency of PSII (Fv/Fm) and operating efficiency of PSII (YII) were calculated as the following equations:(7)FvFm=(Fm−Fo)Fm

(8)YII=(Fm′−Fs)Fm′

YII was decomposed into Fv’/Fm’, PSII operating efficiency of open PSII reaction center, and photochemical quenching (qP), the ratio of open PSII reaction center. qP and non-photochemical quenching (NPQ) were calculated as the following equations: (9)qP=(Fm′−Fs)(Fm′−Fo′)

(10)NPQ=(Fm−Fm′)Fm′

Fo’ is the minimal fluorescence yield in AL and was estimated in the approximation of Oxbrorough and Baker [43]. AL intensity was elevated from low to high levels in a stepwise manner. During the measurements, room temperature was kept at 25 °C.

### 4.4. Measurement of Respiratory O_2_ Uptake Rate

The respiratory O_2_ uptake rates of shoots (11-day HL and 15-day LL plants) and mature leaves (21-day HL and 30-day LL plants) were measured using a liquid-phase Clark-type O_2_ electrode (Rank Brothers, Cambridge, UK) at 23 °C according to Watanabe et al. [37]. Before measuring O_2_ uptake rates, we measured fresh weight (FW) of shoots or leaves and incubated shoots or leaves in buffer (50 mM HEPES, 10 mM MES (pH 6.6), and 0.2 mM CaCl_2_) for 20 min in the dark. The O_2_ uptake rate was calculated assuming that the concentration of O_2_ in the air-saturated buffer was 260 µM at 23 °C. After a constant rate of O_2_ uptake was attained in the buffer, 2 mM KCN was added and the changes in O_2_ uptake rate were analyzed. Preliminary experiments had shown that these concentrations were the most effective. Total respiratory rate (Total R) was estimated as O_2_ uptake rate in the absence of KCN, and CN-resistant O_2_ uptake rate (CN R) was estimated as O_2_ uptake rate in the presence of 2 mM KCN.

### 4.5. Quinone Determination

At 2 h after the onset of the light period, we sampled whole shoots of 11-day or mature leaves of 21-day HL plants. Detailed methods for the extraction of quinones from leaf tissues and the high-performance liquid chromatography (HPLC)-based analysis of quinone contents were described in our previous studies [12,44]. Briefly, the extraction was conducted using freezer-cold acetone after homogenization of leaves in liquid nitrogen. The extraction using acetone was repeated twice, and finally hexane was used to retrieve the residual quinones. The HPLC system was composed as follows: A photodiode array detector (SPD-M20A, Shimadzu, Kyoto, Japan) and a fluorescence detector (RF-10A, Shimadzu) were used. Separation of quinones was carried out by a reverse-phase column (TSKgel ODS-100V, 4.6 × 150 mm, particles 3 µm; Tosoh, Tokyo, Japan) at 40 °C. The flow rate was 1.5 mL·min^–1^ (acetonitrile/ ethanol = 3/1 (*v*/*v*)). We measured oxidized and reduced forms of PQ and UQ and calculated total amount of reduced and oxidized PQ (PQ total), total amount of reduced and oxidized UQ (UQ total), the ratio of reduced form to total amount of PQ (PQ reduction level), and the ratio of reduced form to total amount of UQ (UQ reduction level).

### 4.6. Real-Time PCR

We examined gene expression levels of 2 genes of NDs, *NDA1* and *NDB2*, one gene of UCP, *UCP1*, and one COX gene, *COX6b*. At 2 h after the onset of the light period, we sampled whole shoots of 11-day HL and 15-day LL plants, and measured their FWs. Total RNA was extracted from frozen shoots using TRIzol Reagent (Life Technologies, Tokyo, Japan) according to the manufacturer’s instructions. We used a TURBO DNA-free Kit (Life Technologies) to remove DNA from RNA preparations. RT-PCR was performed using the extracted RNA and a High Capacity RNA-to-cDNA Kit (Life Technologies). Transcript levels were measured using a 7300 Real-time PCR System (Life Technologies). The reaction mixtures contained 1 µL of cDNA, 12.5 µL of 2× Power SYBR Green PCR Master Mix (Life Technologies), 0.5 µL of specific primer (final concentration, 0.2 mM), and 10.5 µL of sterilized water. The PCR conditions were as follows: 50 °C for 2 min, 95 °C for 10 min, and 40 cycles of 95 °C for 15 s, followed by 60 °C for 1 min. Relative transcript levels were calculated by the comparative cycle threshold method. For the internal control, *Actin3* (*ACT3*; At3g53750) was used. The primer sequences of these genes are described in Appendix A.

### 4.7. Metabolome Analysis

At 2 h after the onset of the light period, we sampled whole shoots of 11-day HL and 15-day LL plants, and measured their FWs. Metabolites were extracted from shoots as described previously [14,45,46]. Briefly, frozen shoots were ground with a mortar and pestle in liquid nitrogen and homogenized in 50% (*v*/*v*) methanol (10 µL mg^–1^ of FW) containing 50 µM 1,4-piperazine diethane sulfonic acid and 50 µM methionine sulfone as internal standards. After centrifugation at 21,500× *g* at 4 °C for 5 min, the supernatants were filtered through a 3 kDa cutoff filter (Millipore, Darmstadt, Germany) at 16,100× *g* at 4 °C for 30 min. The filtered samples were analyzed using a capillary electrophoresis system (G1600AX) with a built-in diode-array detector, an MSD mass spectrometer (G1956B), an 1100 series isocratic HPLC pump, a G1603A CE-MS adapter kit, and a G1607A CEESI-MS sprayer kit (all Agilent Technologies, Santa Clara, CA, USA). The amounts of metabolites were corrected against those of internal standards.

### 4.8. Statistical Analysis

Analysis of variance (ANOVA) and Dunnett multiple comparison tests were conducted using R statistical software [47]. In the Dunnett test, we analyzed statistical differences between WT and *pgr5* and between *pgr5* and *aox1a/pgr5*. When normality was not satisfied based on a Shapiro–Wilks test, the data were logarithmically transformed. Analysis of covariance (ANCOVA) was conducted for the difference in *RGR* that was calculated as slopes of linear regression of the natural logarithm of plant dry weight as a function of time. Statistical significance was noted if *p* < 0.1.

## Figures and Tables

**Figure 1 ijms-20-03067-f001:**
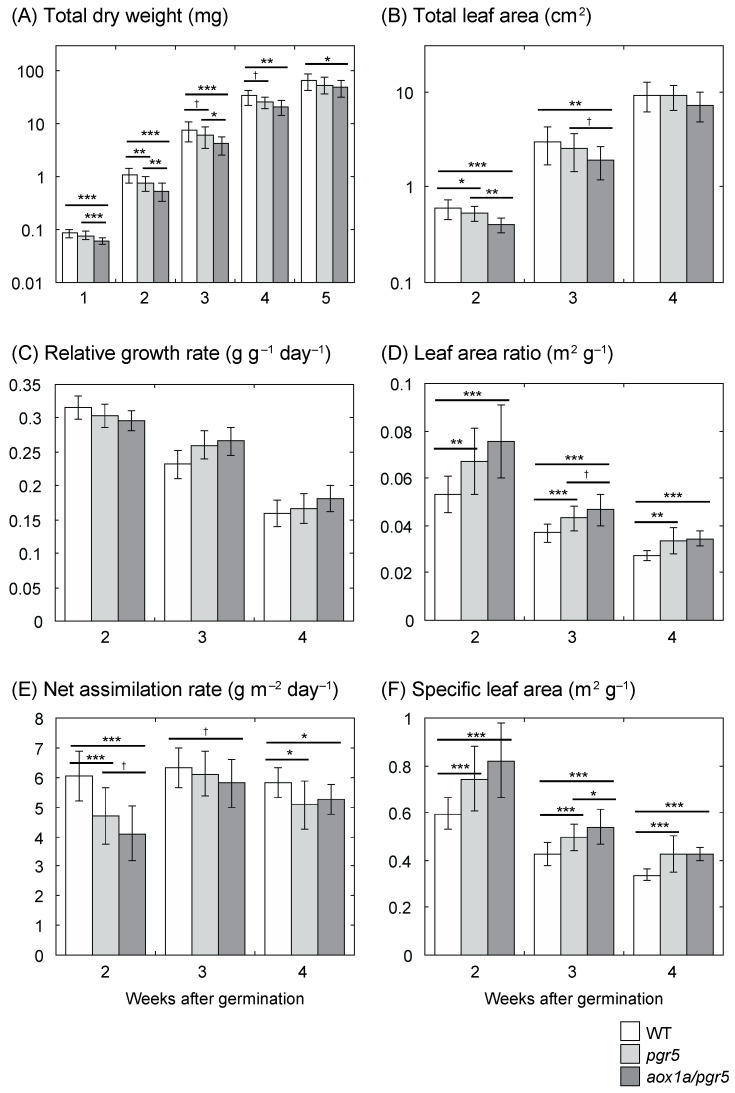
Parameters of growth analysis of high-light (HL) plants. (**A**) Total dry weight of whole plant; (**B**) total leaf area of whole plant; (**C**) relative growth rate (*RGR*); (**D**) leaf area ratio (*LAR*); (**E**) net assimilation rate (*NAR*); (**F**) specific leaf area (*SLA*). Means ± standard deviations are shown except for *RGR* (*n* = 12–24). For data of *RGR*, mean ± 95% confidential interval calculated from regression analysis are shown. Results of one-way ANOVA are shown above three columns, and results of Dunnett multiple comparison test are shown above two columns. † < 0.1, * < 0.05, ** < 0.01, *** < 0.001.

**Figure 2 ijms-20-03067-f002:**
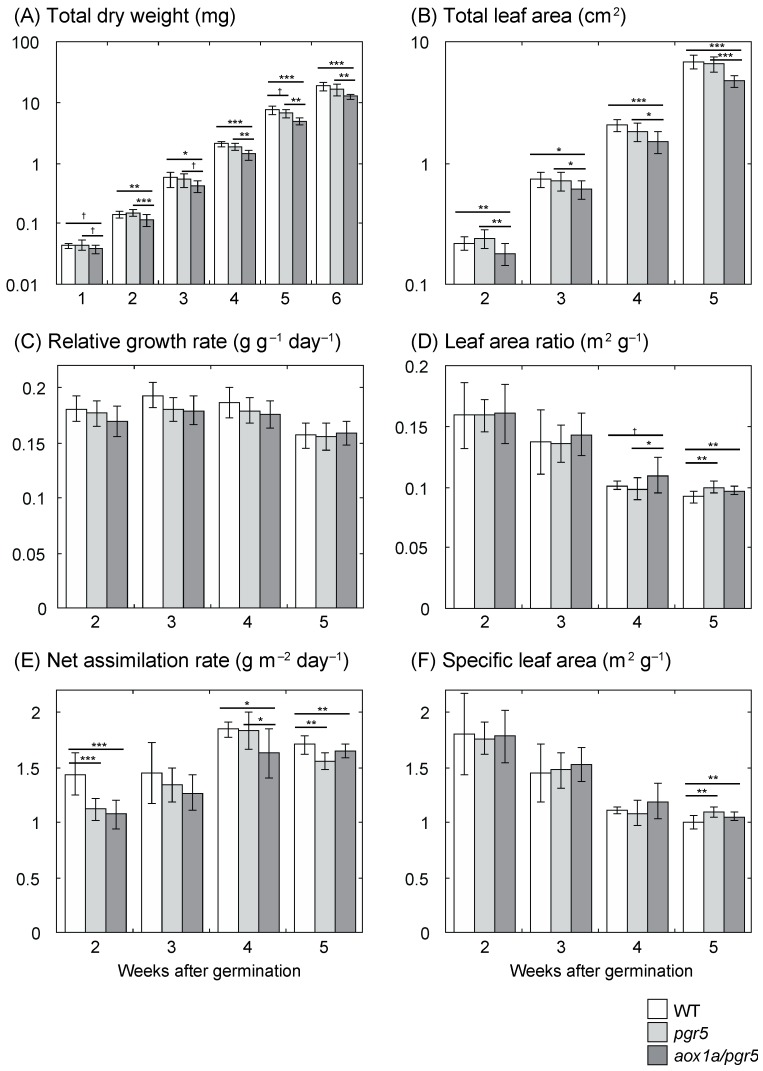
Parameters of growth analysis of low-light (LL) plants. (**A**) Total dry weight of whole plant; (**B**) total leaf area of whole plant; (**C**) relative growth rate (*RGR*); (**D**) leaf area ratio (*LAR*); (**E**) net assimilation rate (*NAR*); (**F**) specific leaf area (*SLA*). Means ± standard deviations are shown except for *RGR* (*n* = 8–12). For data of *RGR*, mean ± 95% confidential interval calculated from the regression analysis are shown. Results of one-way ANOVA are shown above three columns, and results of Dunnett multiple comparison test are shown above two columns. † < 0.1, * < 0.05, ** < 0.01, *** < 0.001.

**Figure 3 ijms-20-03067-f003:**
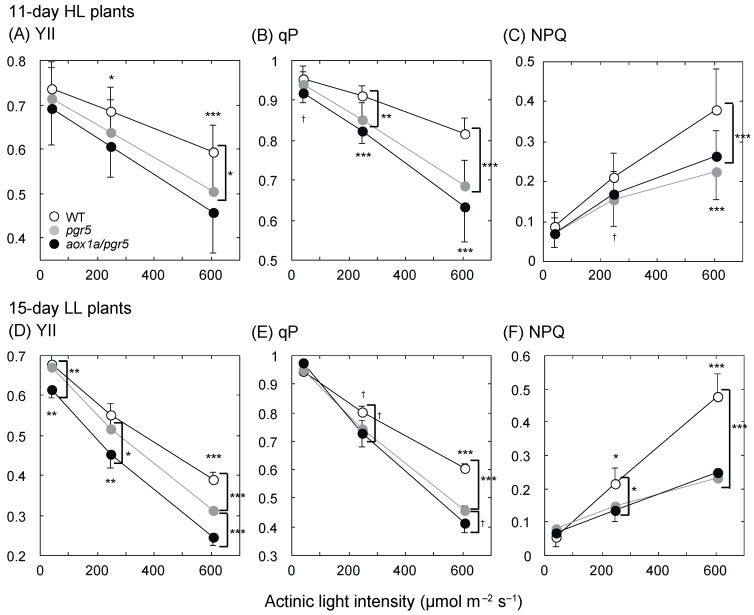
Chlorophyll fluorescence parameters of photosystem II (PSII) of shoots of 11-day high-light (HL) and 15-day low-light (LL) plants. (**A**,**D**) Operating efficiency of PSII (YII); (**B**,**E**) photochemical quenching (qP); (**C**,**F**) non-photochemical quenching (NPQ). (**A**–**C**) HL plants; (**D**–**F**) LL plants. Means ± standard deviations are shown (*n* = 4–12). Results of one-way ANOVA are shown above or below three symbols, and results of Dunnett multiple comparison test are shown near parenthesis. † < 0.1, * < 0.05, ** < 0.01, *** < 0.001.

**Figure 4 ijms-20-03067-f004:**
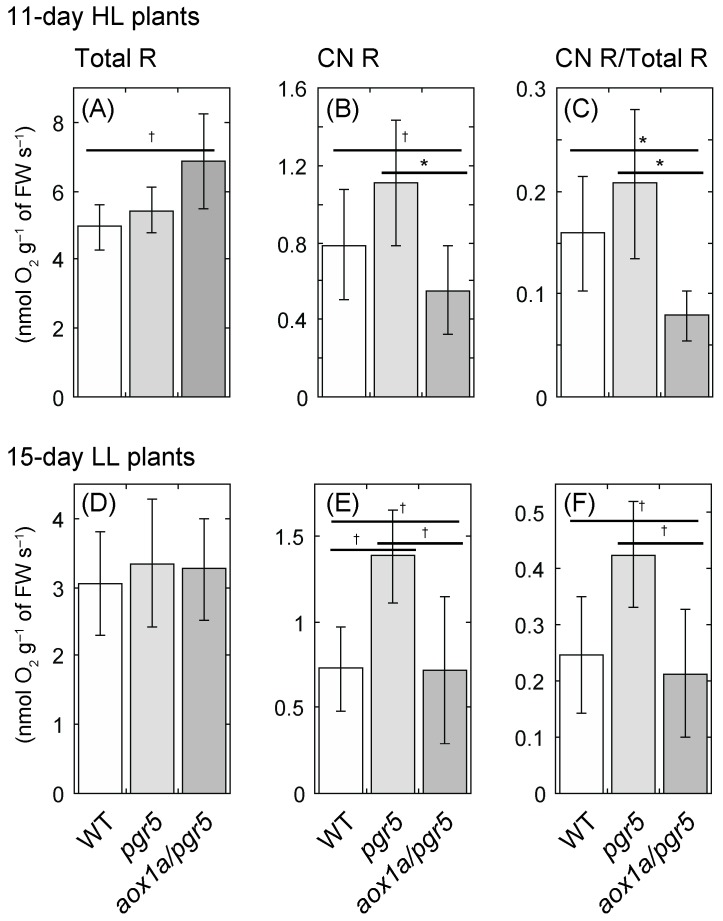
Respiratory O_2_ uptake rate of shoots of 11-day HL and 15-day LL plants. (**A**,**D**) O_2_ uptake rate in the absence of inhibitor (Total R); (**B**,**E**) O_2_ uptake rate in the presence of 2 mM KCN (CN R); (**C,F**) ratio of CN R to total R (CN R/Total R). (**A**–**C**) HL plants; (**D**–**F**) LL plants. Means ± standard deviations are shown (*n* = 3–4). Results of one-way ANOVA are shown above three bars, and results of Dunnett multiple comparison test are shown above two bars. † < 0.1, * < 0.05

**Figure 5 ijms-20-03067-f005:**
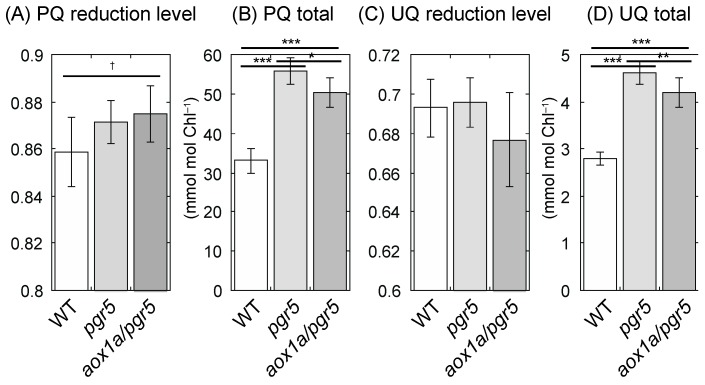
Reduction levels of quinone and total amounts of quinones of shoots of 11-day HL plants. (**A**) PQ reduction level; (**B**) total amount of reduced and oxidized PQ (PQ total); (**C**) UQ reduction level; (**D**) total amount of reduced and oxidized UQ (UQ total). Means ± standard deviations are shown (*n* = 6–8). Results of one-way ANOVA are shown above three bars, and results of Dunnett multiple comparison test are shown above two bars. † < 0.1, * < 0.05, ** < 0.01, *** < 0.001.

**Figure 6 ijms-20-03067-f006:**
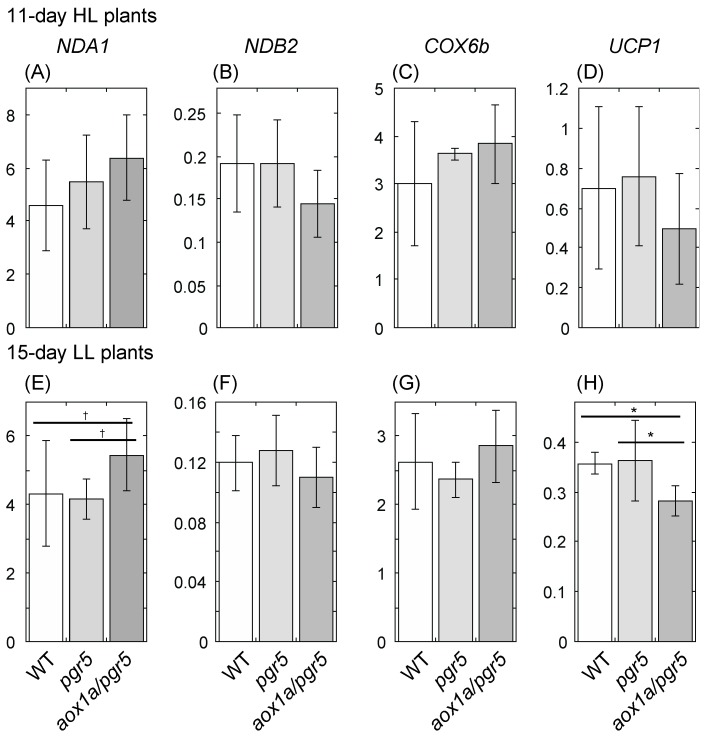
Transcript levels of respiratory genes of shoots of 11-day HL and 15-day LL plants. (**A**,**E**) *NDA1*; (**B**,**F**) *NDB2*; (**C**,**G**) *COX6b*; (**D**,**H**) *UCP1*. (**A**–**D**) HL plants; (**E**–**H**) LL plants. Means ± standard deviations are shown (*n* = 3–7). Results of one-way ANOVA are shown above three bars, and results of Dunnett multiple comparison test are shown above two bars. † < 0.1, * < 0.05.

**Figure 7 ijms-20-03067-f007:**
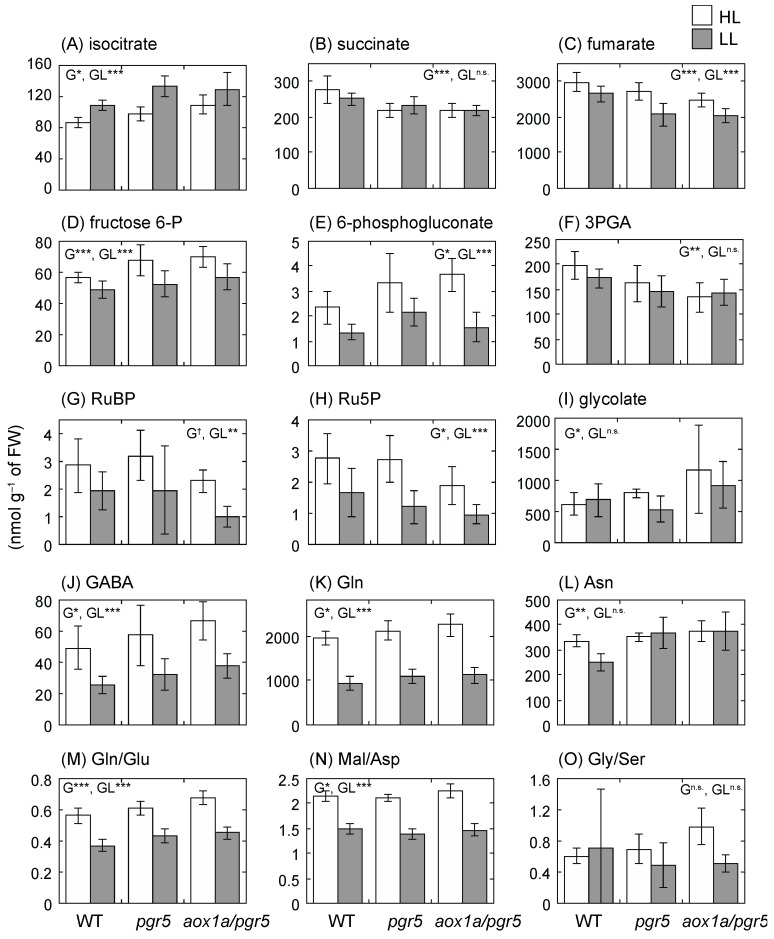
Primary metabolite levels and ratios of shoots of 11-day HL and 15-day LL plants. Means ± standard deviations are shown (*n* = 5). Results of two-way ANOVA are shown in each panel. G and GL denote genotype and growth light factors, respectively, of two-way ANOVA. All probabilities of the interaction of two-way ANOVA are more than 0.05. † < 0.1, * < 0.05, ** < 0.01, *** < 0.001.

**Table 1 ijms-20-03067-t001:** Fv/Fm values of HL and LL plants. Means ± standard deviations are shown (*n* = 4–12). ANOVA *p* denotes probability of one-way ANOVA. * denotes statistical differences between WT and *pgr5* or between *pgr5* and *aox1a/pgr5* using the Dunnett multiple comparison test (*p* < 0.05).

Growth Light Condition	Days after Germination	WT	*pgr5*	*aox1a/pgr5*	ANOVA *p*
HL	11-day	0.786 ± 0.041	0.772 ± 0.053	0.763 ± 0.066	0.602
	21-day	0.791 ± 0.001 *	0.778 ± 0.007	0.770 ± 0.009	0.000996
LL	15-day	0.727 ± 0.014	0.723 ± 0.013	0.647 ± 0.041 *	0.00324
	30-day	0.762 ± 0.010	0.762 ± 0.007	0.761 ± 0.004	0.986

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
