# Peer review of "Mitochondrial AOX Supports Redox Balance of Photosynthetic Electron Transport, Primary Metabolite Balance, and Growth in Arabidopsis thaliana under High Light"

_ijms, 2019, doi:10.3390/ijms20123067_

Reviewer 1 Report

The study “The Mitochondrial Alternative Oxidase Supports Redox Balance of the Photosynthetic Electron Transport, Primary Metabolite Balance and Growth at High Light” is an interesting study but the description of the methods,  the presentation of the results as well as the Discussion and Abstract sections are unsatisfactory so I must recommend to reject the manuscript.

Below I present my objections.

As in Instructions for Authors it is recommended to use abbreviated name for proteins in the title maybe the Authors should change the title replacing the name of Alternative Oxidase by its abbreviation.

I think that the title and particularly Abstract and the description of the aim and results of the study (lines 85-99) should contain the name of the used plant.

Abstract contains too much  background information and too little presentation of the results and conclusions. 

Some results described and discussed in the text were not presented in the manuscript but only in the Supplementary Material. What was the reason? On what base were the parameters presented in the manuscript chosen?

The used abbreviations of the parameters which were not discussed in the text should be explained under Figures – I mean 3PGA, FBP, PEP, DHAP. Maybe it would be better to explain all the used abbreviations in the legends of Figures.

There are inconsistencies in the description of the results. Below the examples are given.

In point 2.4. lines 205-206 Figures 4C and S4C were described as “UQ reduction levels of 11-day and 21-day HL plants were not different among genotypes” as there was no significant differences.  The similar situation was with Figure 5 in point 2.5 lines 224-225.

However, in point 2.1 in description of Figure 1C (lines 107-108) is: “After 3 week, RGR of aox1a/pgr5 was higher than the others” although there is no statistically significant differences. Additionally, Figure 1C lacks SD values so maybe this part of Figure 1 is not complete.

I cannot also agree with the statement (point 2.5., lines 226-227): ”In 11-day HL plants, the COX6b expression level of aox1a/prg5 leaned to be higher than those of WT and pgr5 (Figure 5C) and this trend was similar to that of total R (Fig. 3A).” as in Fig. 3A the differences are much more distinct and significant while in Fig. 5C there is practically no difference between pgr5 and aox1a/pgr5. Additionally the number of samples in Fig. 5 is 4-6, which in my opinion is not enough to make conclusions basing on the above-mentioned results, all the more because SD values are rather high. Similar situation is with Figure S7O and its description in lines 248-249.

There is many results concerning different metabolites in Supplementary Material which are not discussed in the manuscript. In my opinion it would be better to omit these parameters. In contrast those which are discussed in the text should be included into it.

The Authors concluded that (lines 96-99): “The results indicated that aox1a/pgr5 showed slow growth on the early growth stage, in particular in HL, that the difference in growth is ascribed to physiological parameters related to photosynthesis and respiration, and that PSII parameters and balance of primary metabolite levels are affected in aox1a/pgr5.” But why were not PQ and UQ determined in LL plants and why were not gene expression and metabolite levels studied in 21-day HL and 30-day LL-plants, to compare these parameters at early and later stages?

In the Discussion only 11 references from 41 is cited. This section in a considerable part is the repeating of the information included in Results with unnecessary citing Figures and Table.

English should be improved as there are incorrect fragments like:

Line 63 – I think it should be “Unless AOX functions, reactive oxygen species…..”

Lines 67-68 – in my opinion it should be “However, underlying mechanisms affecting changes in metabolic profiles in aoxia1a have been unsolved.

Line 78 – it should be “the study of”

In all the text it should be “at growth stage”.

Line 96 – “in or under conditions”

Line 153 – it should be “similarly”

Line 238 – term not tem.

Line 241 – this sentence lacks than.

Generally, the text should be checked by a native speaker.

Additionally:

INTRODUCTION

1.       Line 50 - I thing e.g., is not necessary by the numbers of the cited references.

2.       Line 53 – it must be [15,16]

3.       Line 60 – after photosystem I(CEF-PSI) was up-regulated” any reference is needed.

MATERIALS AND METHODS

1.       Lines 356-358 - in my opinion the description of RGR does not correlate with the Eqn. 1.

2.       Was the weight used for calculations according to Eqn. 1-6 the dry weight? According the point 4.2 the weighing was performed only once, before drying, and in Results both dry weight and fresh weight are mentioned.

3.       Point 4.3 in my opinion should be extended – the calculated parameters should be named, the text from lines 149-150 should be placed here and the proper equation given, all the more because the full text of reference 36 is not available (no free access).

4.       Line 431 – shouldn’t it be “[14,39,40]”?

5.       Line 433 – I think it would be better “10 ml mg-1  of FW”? Additionally, the abbreviation PIPES should be explained.

6.       Point 4.6. – the Authors should name the studied genes.

7.       Point 4.7- the Authors should name the studied metabolites.

8.       Point 4.8 – with what p value the differences were considered as statistically significant?

9.       Point 4.8 - was the normality of data distribution checked? It is very important because ANOVA can be used only for variables of normal distribution.

RESULTS

1.       Why total dry weight was measured from 1 to 5 week while LA, LAR, SLA only from 2 to 4 week?

2.       Is total dry weight (DW) of whole plant the same as weight of whole plant mentioned in Materials and Methods? Is the variable W mentioned in Materials and Methods the dry or fresh weight of whole plant? It is important because in point 2.3. the authors use the fresh weight described as FW.

3.       Line 106 – according to the authors: “RGR at 2 week was estimated using data from 1 to 3 weeks”. How were RGR estimated at 3 and 4 week – Fig. 1C? Additionally, the way of calculating should be presented not in Results but in Materials and Methods Section, where the description of this part of growth analysis is unclear.

4.       Line 117 –[e.g., 30]. I thing e.g. is not necessary.

5.       Figure 1,2, Table 1 – why n values are so different (n=12-24 or n=4-12 or n=4-12)? 

6.       Line 123-124 - according to authors: “Similarly to HL plants, RGR of aox1a/pgr5 was lower than those of WT and pgr5 (Figure S1C).” It is not true for LL plants at 3 and 5 week as well for HL plants at 3 and 4 week. And why RGR for HL plants was analysed only at 2,3,4 week while for LL ones at 2,3,4,5?

7.       Figure 3 – it should be nmol of O2 g-1 of FW s-1 and Figure 6, S6, S7 - nmol g-1 of FW.

8.       How could the authors explain the differences in Fv/Fm between 11-day HL and 21-day HL  as well as 15-day and 30-day LL?

9.       Point 2.4. – lines 201-205 contain the description of the measured parameters and should be placed in Materials and Methods Section.

10.   Point 2.4 – it lacks the description of PQ and UQ in 11-day HL plants.

11.   Figures 6, S7 and S6 are not clear and must be improved in such way that it is obvious what the  n.s. *** etc. concern.

12.   Line 248 – it should be Figure 6J.

13.   In Figures 6 I, S6P, S6C the vertical axis should be extended.

14.   Figure 6H – I think the abbreviation of the metabolite should be R5P.

DISCUSSION

1.       Line 303 –  it should be [18,34].

2.       Lines 319-320 – the levels of succinate and malate  in HL plants and fumarate in LL ones in pgr5 and aox1a/pgr5 (Fig. 6 B and C, Fig. S6D) seem not to be different.

REFERENCES

1.       Positions 7 and 10 - the abbreviation of the journal should be Trends  Plant Sci.

2.       Positions 23 and 26 and 40 – the number of volume should be given using italics.

3.       Position 38 – there should be a space between an abbreviation of the journal and year.

 Author Response

Reviewer 1

Comments and Suggestions for Authors

The study “The Mitochondrial Alternative Oxidase Supports Redox Balance of the Photosynthetic Electron Transport, Primary Metabolite Balance and Growth at High Light” is an interesting study but the description of the methods, the presentation of the results as well as the Discussion and Abstract sections are unsatisfactory so I must recommend to reject the manuscript.

Authors’ response:Thank you very much for reviewing and commenting to our manuscript. Most of Reviewer 1’s comments can be incorporated in the revised version.

Below I present my objections.

1) As in Instructions for Authors it is recommended to use abbreviated name for proteins in the title maybe the Authors should change the title replacing the name of Alternative Oxidase by its abbreviation. 

Authors’ response:According to the suggestion, I replaced the name of Alternative Oxidase by AOX (see Title).

2) I think that the title and particularly Abstract and the description of the aim and results of the study (lines 85-99) should contain the name of the used plant.

Authors’ response: According to the suggestion, I added the name of the used plant in Abstract (L. 22) and the aims of this study in Introduction (L. 95).

3) Abstract contains too much background information and too little presentation of the results and conclusions.

Authors’ response: According to the suggestion, I reduced the background information and added the results (see Abstract).

4) Some results described and discussed in the text were not presented in the manuscript but only in the Supplementary Material. What was the reason? On what base were the parameters presented in the manuscript chosen?

Authors’ response: Since many figures cannot be presented in the main text and we mentioned that some physiological parameters are affected in aox1a/pgr5of HL plants at the early growth stage, growth parameters of LL plants (Supplementary Figure S1), some data of 21-day HL and 15-day LL plants (Supplementary Figures S2-S4), and many primary metabolite data are presented in the Supplementary Material. However, in the revised version, we moved gene expression data of 15-day LL plants from Supplementary Figure S5 to Figure 5.

5) The used abbreviations of the parameters which were not discussed in the text should be explained under Figures – I mean 3PGA, FBP, PEP, DHAP. Maybe it would be better to explain all the used abbreviations in the legends of Figures.

Authors’ response:We mentioned the results of intermediates of glycolysis and showed full names of these metabolites in the revised version (see L. 275-280).

6) There are inconsistencies in the description of the results. Below the examples are given.

In point 2.4. lines 205-206 Figures 4C and S4C were described as “UQ reduction levels of 11-day and 21-day HL plants were not different among genotypes” as there was no significant differences. The similar situation was with Figure 5 in point 2.5 lines 224-225. However, in point 2.1 in description of Figure 1C (lines 107-108) is: “After 3 week, RGR of aox1a/pgr5 was higher than the others” although there is no statistically significant differences. Additionally, Figure 1C lacks SD values so maybe this part of Figure 1 is not complete.

Authors’ response:According to the suggestion, we conducted statistical analysis for RGRresults using ANCOVA because RGRwas calculated as the slope of a linear regression of the natural logarithm of plant dry weight as a function of time using the data of three sequential samplings (see L. 425-428, L. 525-528). Also, we added the 95% confidential interval for error bars that was calculated from the regression analysis (see figure legends of Figure 1 and Supplementary Figure S1). Since there was not significant difference in RGRamong genotypes, we omitted the description about the difference in RGR (see L. 109-143).

7) I cannot also agree with the statement (point 2.5., lines 226-227): ”In 11-day HL plants, the COX6b expression level of aox1a/prg5 leaned to be higher than those of WT and pgr5 (Figure 5C) and this trend was similar to that of total R (Fig. 3A).” as in Fig. 3A the differences are much more distinct and significant while in Fig. 5C there is practically no difference between pgr5 and aox1a/pgr5. 

Authors’ response:According to the suggestion, we omitted these descriptions (see L. 253-259).

8) Additionally the number of samples in Fig. 5 is 4-6, which in my opinion is not enough to make conclusions basing on the above-mentioned results, all the more because SD values are rather high. Similar situation is with Figure S7O and its description in lines 248-249.

Authors’ response:As the reviewer pointing out, the variances of some data are large, but we cannot add new data to this manuscript now. Based on statistical analysis, we discuss the present results.

9) There is many results concerning different metabolites in Supplementary Material which are not discussed in the manuscript. In my opinion it would be better to omit these parameters. In contrast those which are discussed in the text should be included into it.

Authors’ response:We added some sentences in which many primary metabolites in Results and Discussion sections (L. 275-281, L. 292-294, L. 297-300, L. 393-397). The metabolites that we did not discuss were omitted such as shikimate, cinnamate and ascorbate.

10) The Authors concluded that (lines 96-99): “The results indicated that aox1a/pgr5 showed slow growth on the early growth stage, in particular in HL, that the difference in growth is ascribed to physiological parameters related to photosynthesis and respiration, and that PSII parameters and balance of primary metabolite levels are affected in aox1a/pgr5.” But why were not PQ and UQ determined in LL plants and why were not gene expression and metabolite levels studied in 21-day HL and 30-day LL-plants, to compare these parameters at early and later stages?

Authors’ response:We expected that there is no difference in PQ and UQ reduction level among genotypes because the growth light intensity of LL plants was too low (50 µmol photons m–2 s–1). The qP values at low AL intensity were not different among genotypes (Figure 2E). Since there was no clear difference in total R among genotypes in 21-day HL and 30-day LL plants (Supplementary Figure S3), we did not examine gene expression levels using these plants. Based on a similar reason, we did not examine primary metabolite levels of these plants.

11) In the Discussion only 11 references from 41 is cited. This section in a considerable part is the repeating of the information included in Results with unnecessary citing Figures and Table.

Authors’ response:We added some sentences and references, and omitted the repeating information in Discussion section.

English should be improved as there are incorrect fragments like:

13) Line 63 – I think it should be “Unless AOX functions, reactive oxygen species…..”

Authors’ response:According to the suggestion, we changed it (see L. 69).

14) Lines 67-68 – in my opinion it should be “However, underlying mechanisms affecting changes in metabolic profiles in aoxia1a have been unsolved.

Authors’ response:According to the suggestion, we changed it (see L. 73-74).

15) Line 78 – it should be “the study of”

Authors’ response:According to the suggestion, we changed it (see L. 85).

16) In all the text it should be “at growth stage”.

Authors’ response:According to the suggestion, we changed it (see e.g., L. 32).

17) Line 96 – “in or under conditions”

Authors’ response:According to the suggestion, we changed it (see L. 102).

18) Line 153 – it should be “similarly”

Authors’ response:According to the suggestion, we changed the description (see L. 173).

19) Line 238 – term not tem.

Authors’ response:According to the suggestion, we changed it (see L. 270).

20) Line 241 – this sentence lacks than.

Authors’ response:According to the suggestion, we changed it (see L. 274).

21) Generally, the text should be checked by a native speaker.

Authors’ response:According to the suggestion, the revised version was checked by an English editing service of MDPI.

Additionally:

INTRODUCTION

22) 1.  Line 50 - I thing e.g., is not necessary by the numbers of the cited references.

Authors’ response:According to the suggestion, we changed it (see L. 55).

23) 2.  Line 53 – it must be [15,16]

Authors’ response:According to the suggestion, we changed it (see L. 58).

24) 3.  Line 60 – after photosystem I (CEF-PSI) was up-regulated” any reference is needed.

Authors’ response:According to the suggestion, we changed it (see L. 66).

MATERIALS AND METHODS

25) 1.  Lines 356-358 - in my opinion the description of RGR does not correlate with the Eqn. 1.

Authors’ response:RGRwas estimated as mentioned in L. 425-428. Eqn. (1) was transformed into Eqn. (6), and we estimated NARusing the data of RGRand LAR.

26) 2.  Was the weight used for calculations according to Eqn. 1-6 the dry weight? According the point 4.2 the weighing was performed only once, before drying, and in Results both dry weight and fresh weight are mentioned.

Authors’ response:All data of growth parameters are expressed as DW basis. We mentioned it in L. 427. Respiratory O2 uptake rates and primary metabolite levels are expressed as FW basis (Figures 3, 6, S5 and S6).

27) 3. Point 4.3 in my opinion should be extended – the calculated parameters should be named, the text from lines 149-150 should be placed here and the proper equation given, all the more because the full text of reference 36 is not available (no free access).

Authors’ response:According to the suggestion, we added sentences (see L. 459-463).

28) 4.  Line 431 – shouldn’t it be “[14,39,40]”?

Authors’ response:According to the suggestion, we added sentences (see L. 512).

29) 5.  Line 433 – I think it would be better “10 ml mg-1 of FW”? Additionally, the abbreviation PIPES should be explained.

Authors’ response:According to the suggestion, we added sentences (see L. 514).

30) 6.  Point 4.6. – the Authors should name the studied genes.

Authors’ response:According to the suggestion, we added sentences (see L. 496-497).

31) 7.  Point 4.7- the Authors should name the studied metabolites.

Authors’ response:We measured many primary metabolites, and thus we did not write the names in Materials and Methods.

32) 8. Point 4.8 – with what p value the differences were considered as statistically significant?

Authors’ response:We considered the significant difference when p< 0.1. We mentioned it in Materials and Methods (see L. 528-529).

33) 9. Point 4.8 - was the normality of data distribution checked? It is very important because ANOVA can be used only for variables of normal distribution.

Authors’ response:According to the suggestion, we checked the normality of data distribution. When normality was not satisfied based on a Shapiro-Wilks test, the data were logarithmically transformed. We mentioned it in Materials and Methods (see L. 525-526).

RESULTS

34) 1. Why total dry weight was measured from 1 to 5 week while LA, LAR, SLA only from 2 to 4 week?

Authors’ response:We estimated leaf area from 2 to 4 weeks (Figure 2B), and so we calculated LAR and SLA from 2 to 4 weeks. Since we estimated RGR from 2 to 4 weeks, total dry weight was measured from 1 to 5 weeks.

35) 2. Is total dry weight (DW) of whole plant the same as weight of whole plant mentioned in Materials and Methods? Is the variable W mentioned in Materials and Methods the dry or fresh weight of whole plant? It is important because in point 2.3. the authors use the fresh weight described as FW.

Authors’ response:W is dry weight of whole plant in equations.

36) 3.  Line 106 – according to the authors: “RGR at 2 week was estimated using data from 1 to 3 weeks”. How were RGR estimated at 3 and 4 week – Fig. 1C? Additionally, the way of calculating should be presented not in Results but in Materials and Methods Section, where the description of this part of growth analysis is unclear.

Authors’ response:As we mentioned in Materials and Methods (L. 425-428), RGR was calculated as the slope of a linear regression of the natural logarithm of plant dry weight as a function of time using the data of three sequential samplings. RGR at 3 week was estimated using data from 2 to 4 weeks. 

37) 4.  Line 117 –[e.g., 30]. I thing e.g. is not necessary.

Authors’ response:According to the suggestion, we omitted it (see L. 128).

38) 5. Figure 1,2, Table 1 – why n values are so different (n=12-24 or n=4-12 or n=4-12)? 

Authors’ response:We used large number of samples of growth parameter data at 1, 2 and 3 weeks and photosynthetic parameters of 11-day HL plants.

39) 6. Line 123-124 - according to authors: “Similarly to HL plants, RGR of aox1a/pgr5 was lower than those of WT and pgr5 (Figure S1C).” It is not true for LL plants at 3 and 5 week as well for HL plants at 3 and 4 week. And why RGR for HL plants was analysed only at 2,3,4 week while for LL ones at 2,3,4,5?

Authors’ response:Growth rate of LL plants was lower than that of HL plants, and so we examined data of one more week for LL plants.

40) 7. Figure 3 – it should be nmol of O2 g-1 of FW s-1 and Figure 6, S6, S7 - nmol g-1 of FW.

Authors’ response:According to the suggestion, we changed them (see Figures 6, S5 and S6).

41) 8. How could the authors explain the differences in Fv/Fm between 11-day HL and 21-day HL  as well as 15-day and 30-day LL?

Authors’ response:According to the suggestion, we changed them (see Figures 6, S5 and S6).

42) 9. Point 2.4. – lines 201-205 contain the description of the measured parameters and should be placed in Materials and Methods Section.

Authors’ response:According to the suggestion, the description was placed in Materials and Methods (see L. 489-493).

43) 10. Point 2.4 – it lacks the description of PQ and UQ in 11-day HL plants.

Authors’ response:According to the suggestion, the description of PQ and UQ was mentioned (see L. 234-236).

44) 11. Figures 6, S7 and S6 are not clear and must be improved in such way that it is obvious what the n.s. *** etc. concern.

Authors’ response:According to the suggestion, we changed the expression of ANOVA results (see Figures 6, S5, and S6).

45) 12. Line 248 – it should be Figure 6J.

Authors’ response:According to the suggestion, we changed it (see L. 287).

46) 13. In Figures 6 I, S6P, S6C the vertical axis should be extended.

Authors’ response:According to the suggestion, the vertical axis was changed (see Figures 6I, S5C). Figure S6P was omitted.

47) 14. Figure 6H – I think the abbreviation of the metabolite should be R5P.

Authors’ response:This metabolite is Ru5P. We changed our mistake in description of Results (L. 303-304).

DISCUSSION

48) 1. Line 303 –  it should be [18,34].

Authors’ response:According to the suggestion, we changed it (see L. 358).

49) 2. Lines 319-320 – the levels of succinate and malate in HL plants and fumarate in LL ones in pgr5 and aox1a/pgr5 (Fig. 6 B and C, Fig. S6D) seem not to be different.

Authors’ response:According to the suggestion, we changed the description (see L. 374-375).

REFERENCES

50) 1.  Positions 7 and 10 - the abbreviation of the journal should be Trends Plant Sci.

Authors’ response:According to the suggestion, we changed it (see Reference No. 5). Reference No. 10 was omitted.

51) 2.  Positions 23 and 26 and 40 – the number of volume should be given using italics.

Authors’ response:According to the suggestion, we changed them (see Reference No. 20, 23, 40).

52) 3. Position 38 – there should be a space between an abbreviation of the journal and year.

Authors’ response:According to the suggestion, we changed them (see Reference No. 38).

Reviewer 2 Report

The current version of the paper is well presented and structured and all the experiments have been carried out properly and the data analyzed and interpreted as expected. Considering these premises, I recommend the paper for publication after a language revision.

In this study,  the authors studied the mitochondrial AOX efficiently function as oxidation of excess reductants that accumulate at High Light comparing pgr5 and photosynthetic mutant aox1a/pgr5. They have measured and compared physiological and biochemical traits of wild type and mutant plants and integrated the results with metabolic and molecular analysis. The authors have used a good methodologic approach and they have obtained important results able to characterize the mitochondrial AOX functionality and to lay the foundation for the study of underlying mechanisms of these phenomena.

Considering these premises, I recommend the paper for publication after a language revision.

I suggest a minor use of that, those, these and this, especially in the result section.

Author Response

Reviewer 2

Comments and Suggestions for Authors

The current version of the paper is well presented and structured and all the experiments have been carried out properly and the data analyzed and interpreted as expected. Considering these premises, I recommend the paper for publication after a language revision. 

In this study, the authors studied the mitochondrial AOX efficiently function as oxidation of excess reductants that accumulate at High Light comparing pgr5 and photosynthetic mutant aox1a/pgr5. They have measured and compared physiological and biochemical traits of wild type and mutant plants and integrated the results with metabolic and molecular analysis. The authors have used a good methodologic approach and they have obtained important results able to characterize the mitochondrial AOX functionality and to lay the foundation for the study of underlying mechanisms of these phenomena. 

53) Considering these premises, I recommend the paper for publication after a language revision. 

I suggest a minor use of that, those, these and this, especially in the result section.

Authors’ response:Thank you very much for reviewing and commenting to our manuscript.According to the suggestion, the revised version was checked by an English editing service of MDPI.

Round  2

Reviewer 1 Report

Dear Authors,

The study The Mitochondrial AOX Supports Redox Balance of the Photosynthetic ElectronTransport, Primary Metabolite Balance, and Growth in Arabidopsis thaliana under High Light

by Zhenxiang Jiang, Chihiro K.A. Watanabe, Atsuko Miyagi, Maki Kawai-Yamada, Ichiro Terashima , Ko Noguchi is an interesting study. The Authors performed a considerable amount of research and obtained interesting results. However, their presentation is unacceptable, particularly having regarded that the main objection presented in my previous review were not taken into account in a satisfactory way.

Many data were not presented in the text. The Authors suggested in their response to point 4 that they focused on the material essential for the study. However, a considerable quantity of results not included in the manuscript is not only described in Results section but also discussed in the Discussion. I think that it a wrong way of presenting of the obtained results not to describe them in Results section with a detailed description in Discussion section (malate, 2-OG, Gln/Glu, Gly/Ser, Mal/Asp), all the more so because they consist the base for the conclusions. Generally, Discussion is rather poor and in a considerable part it is the repeating of the Result section or its supplement.

Again only about 1/3 of the references (15 of 41) were cited in this Section.

Certainly the comprehension of the study would be better if all the results, at least those discussed in the text, were placed in it. If it is impossible to present so many figures in the manuscript maybe the Authors should present some part of the results in form of Tables.

The answer concerning point 25 from my first review is not satisfactory – the Authors stated “Relative growth rate (RGR, g g–1 day–1) was calculated as the slope of a linear regression of the natural logarithm of plant dry weight as a function of time using the data of 3three sequential samplings.” In Eqn. (1) there is no logarithm.

Point 33 – I still do not understand why when normality was not satisfied according to Shapiro-Wilk test, the data were logarithmically transformed instead of using the statistical tests for non-normal distribution.

In the first version there was no information about p values with which the differences were considered to be significant. In the new version the Authors give the value 0.1 but in biological experiments it should be 0.05.

The answer to point 41 is not connected with it.

The description of the results contains some mistakes or lack of clarity.

 ABSTRACT

1.       Lines 32-33  - the description of leaf area ratio results is unclear.

INTRODUCTION

1.       Lines 63-66 – “Under moderate drought stress, the electron transport of PSII was lowered and protons accumulated in the thylakoid lumen in AOX-knock-down (AOX-KD) tobacco leaves [169,2017]. In the AOX-KD tobacco, the operating efficiency of PSI (YI) was not inhibited and the cyclic electron flow around photosystem I (CEF-PSI) was up-regulated [16,17].” In this fragment literature references 16,17 should be given once, after the second sentence.

2.       Lines 71, 79 – the numbers of the reference positions are wrong [114], [214].

RESULTS

1.       Figure 5 – what was the number of samples? Why was it changed while the Authors stated that “we cannot add new data to this manuscript now”?

2.       Point 38 from the first review – the answer is not relevant and satisfactory.

3.       Lines 111-113 – the Authors stated that “these values of aox1a/pgr5 were significantly lower than those of WT and pgr5 “ but the comparison of WT and aox1a/pgr5 was not performed (I mean Dunnet test). Additionally, the value p=0.0713 should be explained - aox1a/pgr5 vs. pgr5

4.        Line 114-115 – the description of RGR calculation is in Materials and Methods and it is not necessarily to repeat it in Results.

5.       Line 115-116 - if in Materials and Methods the Authors stated that statistical significance was noted if p < 0.1 in this sentence it should be p > 0.1.

6.       Line 118-119 – p=0.0700 concerns the difference between aox1a/pgr5 and pgr5, not aox1a/pgr5 and the others. Additionally it is not the only significant effect in case of LAR.

7.       Line 127 – p=0.0612 concerns the difference between aox1a/pgr5 and pgr5, not aox1a/pgr5 and the others.

8.       I think that Figure S1 should be included into text as there is a detailed description of it in the Results section. Generally, there is a lot of results which are described in the text and not presented. The Authors answered that it was impossible to place all the results in the text but maybe these data could be given in form of Tables which would take much less space.

9.       Line 140-142 – NAR of aox1a/pgr5 was lower after 2 and 3 weeks.

10.   Line 167-168 – this sentence should be removed into point 4.3 and the proper equations should be given for all flurescence parameters like in point 4.2.

11.   Lines 180-181 – the Authors stated: “The qP of aox1a/pgr5 was lower than that those of the other genotypes (p = 0.0551;, Figure 2E). This p value concerns the difference between aox1a/pgr5 and pgr5, not aox1a/pgr5 and the others.

12.   Line 201 – fresh weigh was used so in Materials and Methods the measure of plant weight before drying should be mentioned.

13.   Line 238-239 – this sentence should be improved.

14.   Line 278 – phosphate.

15.   Figure 6L and line 291 - Asn in aox1a/pgr5 and pgr5 seem to be the same or very similar, not different.

16.   Line 297 – I think it should be aspartate and what does it mean Asp, lysine (Lys).

17.   Lines 297-299 – Lys is not different among the studied genotypes.

18.   Figure 6E and line 303 – 6-phosphogluconate in  aox1a/pgr5 is not higher than in pgr5.

DISCUSSION

1.       Citing the Figures in this Section seems to be unnecessary.

2.       Lines 346-349 – “In AOX-KD tobacco leaves underat moderate drought stress, an decreased amount of the chloroplast proton-ATP synthase decreased, proton accumulated in the thylakoid lumen, the electron transport in the cytochrome b6f complex was inhibited, and the electron transport of PSII was also inhibited [16,1720]. In those the AOX-KD tobacco leaves, CEF-PSI was up-regulated and high NPQ was induced.”

It seems that the references should be in the end of this fragment - compare Introduction lines 63-66.

3.       Line 370 [114]?

4.       Line 383 “6-phosohogluconate” ?

5.       Lines 374-377 of Discussion and 281-283 of Results contain practically the same information with addition of malate results in Discussion.

6.       Lines 384-388 - “Although levels of glycine (Gly) and serine (Ser), and the ratio of Gly to Ser (Gly/Ser) were not different among genotypes (Figure 6O, Supplementary Figure S6A,B), the ratio of malate to aspartate (Mal/Asp) in aox1a/pgr5 was higher than in the others (Figure 6N), suggesting that the flux of Mal/Asp shuttle between the cytosol and mitochondrion may be changed in aox1a/pgr5 [2].” I cannot understand why any reference is given here. It seems to me that it is a conclusion drawn from this study.

MATERIALS AND METHODS

1.       According to the description of the calculation of RGR in Eqn.1 any logarithm should be.

2.       Line 436 – it should be explained that W is dry weight. In point 2.1. the abbreviation of dry weight is DW so I think that it should be used in equations 1, 2 3, 5. Additionally, is the LW weight of dry or fresh leaves?

 Author Response

Reviewer 1

Comments and Suggestions for Authors

The study The Mitochondrial AOX Supports Redox Balance of the Photosynthetic ElectronTransport, Primary Metabolite Balance, and Growth in Arabidopsis thaliana under High Light by Zhenxiang Jiang, Chihiro K.A. Watanabe, Atsuko Miyagi, Maki Kawai-Yamada, Ichiro Terashima, Ko Noguchi is an interesting study. The Authors performed a considerable amount of research and obtained interesting results. However, their presentation is unacceptable, particularly having regarded that the main objection presented in my previous review were not taken into account in a satisfactory way.

Authors’ response:Thank you very much for reviewing and commenting to our manuscript. Most of Reviewer 1’s comments can be incorporated in the new version.

1)     Many data were not presented in the text. The Authors suggested in their response to point 4 that they focused on the material essential for the study. However, a considerable quantity of results not included in the manuscript is not only described in Results section but also discussed in the Discussion. I think that it a wrong way of presenting of the obtained results not to describe them in Results section with a detailed description in Discussion section (malate, 2-OG, Gln/Glu, Gly/Ser, Mal/Asp), all the more so because they consist the base for the conclusions.

Authors’ response: According to the suggestion, we moved the sentences about results of metabolite ratios from Discussion into Results section (see L. 281-283, L. 389-392).

2)     Generally, Discussion is rather poor and in a considerable part it is the repeating of the Result section or its supplement. Again only about 1/3 of the references (15 of 41) were cited in this Section.

Authors’ response: According to the suggestion, we deleted the repeating of the Result section, and added some discussion and references in Discussion section (see L. 298-402).

3)     Certainly the comprehension of the study would be better if all the results, at least those discussed in the text, were placed in it. If it is impossible to present so many figures in the manuscript maybe the Authors should present some part of the results in form of Tables.

Authors’ response: According to the suggestion, we moved the results of growth analysis of LL plants into the main text as Figure 2 (see Figure 2).

4)     The answer concerning point 25 from my first review is not satisfactory – the Authors stated “Relative growth rate (RGR, g g–1 day–1) was calculated as the slope of a linear regression of the natural logarithm of plant dry weight as a function of time using the data of 3three sequential samplings.” In Eqn. (1) there is no logarithm.

Authors’ response:We did not use Eqn. (1) to calculate RGR. RGR was calculated as the slope of a linear regression of the natural logarithm of plant dry weight using the data of three sequential samplings. To estimate NAR, we used Eqn. (6) that is transformed equation of Eqn. (1). We changed the expression to understand the calculations easily (see L. 428-450).

5)     Point 33 – I still do not understand why when normality was not satisfied according to Shapiro-Wilk test, the data were logarithmically transformed instead of using the statistical tests for non-normal distribution.

Authors’ response:We used parametric tests using logarithmically transformed data because parametric tests show high ability to detect the statistical difference compared to non-parametric tests.

6)     In the first version there was no information about p values with which the differences were considered to be significant. In the new version the Authors give the value 0.1 but in biological experiments it should be 0.05.

Authors’ response:In many biological experiments, the probability 0.05 is often used. However the probability 0.1 is accepted, provided that the probability is noted (e.g., Noguchi et al. Plant Cell Physiol, 59, 637–649, 2018). Therefore we mentioned that “statistical significance was noted if p< 0.1” in the text (see L. 544).

7)     The answer to point 41 is not connected with it.

Authors’ response:We are very sorry not to mention it properly. According to the suggestion, we added the discussion on it (see L. 334-339).

8)     The description of the results contains some mistakes or lack of clarity.

Authors’ response:We are very sorry not to correct all mistakes in the previous version. We changed them carefully in this version.

ABSTRACT

9)     1. Lines 32-33  - the description of leaf area ratio results is unclear.

Authors’ response: According to the suggestion, we changed the expression (see L. 28).

INTRODUCTION

10)   1. Lines 63-66 – “Under moderate drought stress, the electron transport of PSII was lowered and protons accumulated in the thylakoid lumen in AOX-knock-down (AOX-KD) tobacco leaves [169,2017]. In the AOX-KD tobacco, the operating efficiency of PSI (YI) was not inhibited and the cyclic electron flow around photosystem I (CEF-PSI) was up-regulated [16,17].” In this fragment literature references 16,17 should be given once, after the second sentence.

Authors’ response: According to the suggestion, we changed the expression (see L. 58-61).

11)   2. Lines 71, 79 – the numbers of the reference positions are wrong [114], [214].

Authors’ response: According to the suggestion, we changed the position of references [11,18] (see L. 68). However, we did not change the position of reference [21] (see L. 74). We think it is not wrong.

RESULTS

12)   1. Figure 5 – what was the number of samples? Why was it changed while the Authors stated that “we cannot add new data to this manuscript now”?

Authors’ response: When we made the previous version, we combined the data of 11-day HL plants with those of 15-day LL plants for Figure 5 (See the file of the previous version, Jiang et al 2019_IJMS_R1.docx). Therefore the number of data was changed.

13)   2. Point 38 from the first review – the answer is not relevant and satisfactory.

Authors’ response: We used large number of samples at the early growth stage, and small number of samples at the later stages. Therefore the range of numbers is varied.

14)   3. Lines 111-113 – the Authors stated that “these values of aox1a/pgr5 were significantly lower than those of WT and pgr5 “ but the comparison of WT and aox1a/pgr5 was not performed (I mean Dunnet test). Additionally, the value p=0.0713 should be explained - aox1a/pgr5 vs. pgr5

Authors’ response: According to the suggestion, we changed the expression (see L. 106).

15)   4. Line 114-115 – the description of RGR calculation is in Materials and Methods and it is not necessarily to repeat it in Results.

Authors’ response: According to the suggestion, we deleted the description (see L. 108).

16)   5. Line 115-116 - if in Materials and Methods the Authors stated that statistical significance was noted if p < 0.1 in this sentence it should be p > 0.1.

Authors’ response: According to the suggestion, we changed the expression (see L. 109).

17)   6. Line 118-119 – p=0.0700 concerns the difference between aox1a/pgr5 and pgr5, not aox1a/pgr5 and the others. Additionally it is not the only significant effect in case of LAR.

Authors’ response: According to the suggestion, we changed the expression (see L. 112).

18)   7. Line 127 – p=0.0612 concerns the difference between aox1a/pgr5 and pgr5, not aox1a/pgr5 and the others.

Authors’ response: According to the suggestion, we changed the expression (see L. 117).

19)   8. I think that Figure S1 should be included into text as there is a detailed description of it in the Results section. Generally, there is a lot of results which are described in the text and not presented. The Authors answered that it was impossible to place all the results in the text but maybe these data could be given in form of Tables which would take much less space.

Authors’ response: According to the suggestion, we moved the results of growth analysis of LL plants into the main text as Figure 2 (see Figure 2).

20)   9. Line 140-142 – NAR of aox1a/pgr5 was lower after 2 and 3 weeks.

Authors’ response: According to the suggestion, we changed the expression (see L. 130).

21)   10. Line 167-168 – this sentence should be removed into point 4.3 and the proper equations should be given for all flurescence parameters like in point 4.2.

Authors’ response: According to the suggestion, we moved the sentence to 4.3. and added the equations (see L. 464-476).

22)   11. Lines 180-181 – the Authors stated: “The qP of aox1a/pgr5 was lower than that those of the other genotypes (p = 0.0551;, Figure 2E). This p value concerns the difference between aox1a/pgr5 and pgr5, not aox1a/pgr5 and the others.

Authors’ response: According to the suggestion, we changed the expression (see L. 174).

23)   12. Line 201 – fresh weigh was used so in Materials and Methods the measure of plant weight before drying should be mentioned.

Authors’ response: According to the suggestion, we added sentences (see L. 489-490, 513, 527).

24)   13. Line 238-239 – this sentence should be improved.

Authors’ response: According to the suggestion, we changed the expression (see L. 223-224).

25)   14. Line 278 – phosphate.

Authors’ response: According to the suggestion, we changed the expression (see L. 258).

26)   15. Figure 6L and line 291 - Asn in aox1a/pgr5 and pgr5 seem to be the same or very similar, not different.

Authors’ response: According to the suggestion, we changed the expression (see L. 269-273).

27)   16. Line 297 – I think it should be aspartate and what does it mean Asp, lysine (Lys).

Authors’ response: According to the suggestion, we changed the expression (see L. 278-280).

28)   17. Lines 297-299 – Lys is not different among the studied genotypes.

Authors’ response: According to the suggestion, we changed the expression (see L. 279).

29)   18. Figure 6E and line 303 – 6-phosphogluconate in aox1a/pgr5 is not higher than in pgr5.

Authors’ response: According to the suggestion, we changed the expression (see L. 284-287).

DISCUSSION

30)   1. Citing the Figures in this Section seems to be unnecessary.

Authors’ response: According to the suggestion, we deleted unnecessary citations in Discussion (see Discussion section).

31)   2. Lines 346-349 – “In AOX-KD tobacco leaves under at moderate drought stress, an decreased amount of the chloroplast proton-ATP synthase decreased, proton accumulated in the thylakoid lumen, the electron transport in the cytochrome b6f complex was inhibited, and the electron transport of PSII was also inhibited [16,1720]. In those the AOX-KD tobacco leaves, CEF-PSI was up-regulated and high NPQ was induced.” It seems that the references should be in the end of this fragment - compare Introduction lines 63-66.

Authors’ response: According to the suggestion, we changed the expression (see L. 323-328).

32)   3. Line 370 [114]?

Authors’ response:This reference is [11] (see L. 376).

33)   4. Line 383 “6-phosohogluconate” ?

Authors’ response:According to the suggestion, we changed the word (see L. 388).

34)   5. Lines 374-377 of Discussion and 281-283 of Results contain practically the same information with addition of malate results in Discussion.

Authors’ response: According to the suggestion, we changed the expression (see L. 378-382).

35)   6. Lines 384-388 - “Although levels of glycine (Gly) and serine (Ser), and the ratio of Gly to Ser (Gly/Ser) were not different among genotypes (Figure 6O, Supplementary Figure S6A,B), the ratio of malate to aspartate (Mal/Asp) in aox1a/pgr5 was higher than in the others (Figure 6N), suggesting that the flux of Mal/Asp shuttle between the cytosol and mitochondrion may be changed in aox1a/pgr5 [2].” I cannot understand why any reference is given here. It seems to me that it is a conclusion drawn from this study.

Authors’ response: According to the suggestion, we changed the expression (see L. 392-393).

MATERIALS AND METHODS

36)   1. According to the description of the calculation of RGR in Eqn.1 any logarithm should be.

Authors’ response:As we mentioned it in Point 4, we did not use Eqn. (1) to calculate RGR. RGR was calculated as the slope of a linear regression of the natural logarithm of plant dry weight using the data of three sequential samplings. To estimate NAR, we used Eqn. (6) that is transformed equation of Eqn. (1). We changed the expression to understand the calculations easily (see L. 428-450).

37)   2. Line 436 – it should be explained that W is dry weight. In point 2.1. the abbreviation of dry weight is DW so I think that it should be used in equations 1, 2 3, 5. Additionally, is the LW weight of dry or fresh leaves?

Authors’ response: W and LW are dry weights of whole plant and leaves, respectively. We added “dry” in the sentences.

Round  3

Reviewer 1 Report

The new version of the article “Mitochondrial AOX Supports Redox Balance of Photosynthetic Electron Transport, Primary Metabolite Balance, and Growth in Arabidopsis thaliana under High Light” by Zhenxiang Jiang, Chihiro K.A. Watanabe, Atsuko Miyagi, Maki Kawai-Yamada, Ichiro Terashima, Ko Noguchi in my opinion has not been improved enough to warrant publication in IJMS.

My main objections concerns statistics. In the answer (5) the Authors stated “We used parametric tests using logarithmically transformed data because parametric tests show high ability to detect the statistical difference compared to non-parametric tests.” In my opinion there is no basis for such an analysis. In case of non-normal distribution the proper tests should be used. Maybe this is why sometimes the statistical significance shown by the Authors can hardly be observed.

In my comments I suggested that some data should be given in form of Tables. Surely in such case it would be better to observe the differences, because in the current form of the article some statements are difficult to accept, for instance in lines 270-272 there is: “A level of glutamine (Gln), which has a high nitrogen (N)/carbon (C) ratio, was significantly different among genotypes, and the level in aox1a/pgr5 was higher than in the other genotypes (Figure 7K).” Looking at Figure 7K it can hardly be observed in case of LL plants. A similar situation is in case of Thr, Gln/Glu ans Mal/Asp. Mal/Asp in aox1a/pgr5 does not seem to be significantly higher than in WT.

In Figures 7, S4, S5 “G” denotes genotype but it is not clear if it concerns LL or HL plants and in some cases the differences do not seem to be the same e.g.: Asn - Fig. 7(L), Thr  Fig. S5(S), 3PGA Fig. 6(F).

There are inconsistencies concerning statistical significance – when the differences are significant - with p<0.05 (line 107, 124) or p<0.1 (line 545)?

I am not satisfied with the answer for point 13 – the number of samples should be the same or at least similar to enable an accurate comparison, particularly in view of conclusion 2.

Some part of Discussion (points 3.1 and 3.2) have been considerably improved. The repeating of results was removed and the discussion itself was enriched. However, in lines 368, 381 and 402-403 the repeating of the results with citing Figures has been left. Point 3.3 has not been enriched compared to the previous version.

There is still inconsistency concerning RGR – if Eqn. (1) was not used for RGR calculation why then RGR was defined with using Eqn. (1).

Below I present some other objections.

MATERIALS AND METHODS

1.      I still cannot understand why in point 4.2 W is the dry weight of whole plant and in point 2.1 DW (the total dry weight) is used, a term which has not been not explained in Materials and Methods part.

2.      Point 4.4. line 491 I think it should be”measurements”

3.      The sentence about measuring the fresh weight of shoots or leaves should be placed at the beginning of the 4.4. point since in the next sentence “these concentrations” are mentioned and in this situation it is not clear what are “these concentrations”.

RESULTS

1.      Lines 116-117 “SLA of aox1/pgr5 was higher than that of pgr5”. It is true only after 3 weeks.

2.      The sentence from lines 167-168 from the previous version was placed in point 4.3  but was not removed from point 2.2.

3.      Table 1 – superscript letters suggest that in 15-day LL plants and 21-day HL plants Fv/Fm values for WT and aox1a/pgr5 are not statistically different according Dunnett test while such comparison was not performed. The letters should be changed.

4.      Line 241 p=0.0603 for aox1a/pgr5 – is it a result of ANOVA or Dunnett test – I mean vs. what?

5.      Line 260 “phosphate”.

6.      Line 279 – “aspartate”.